# Multi-Task Transformer with Adaptive Cross-Entropy Loss for Multi-Dialect Speech Recognition

**DOI:** 10.3390/e24101429

**Published:** 2022-10-08

**Authors:** Zhengjia Dan, Yue Zhao, Xiaojun Bi, Licheng Wu, Qiang Ji

**Affiliations:** 1School of Information Engineering, Minzu University of China, Beijing 100081, China; 2Department of Electrical, Computer, and Systems Engineering, Rensselaer Polytechnic Institute, Troy, NY 12180-3590, USA

**Keywords:** adaptive cross-entropy loss, multi-task Transformer, multi-dialect speech recognition

## Abstract

At present, most multi-dialect speech recognition models are based on a hard-parameter-sharing multi-task structure, which makes it difficult to reveal how one task contributes to others. In addition, in order to balance multi-task learning, the weights of the multi-task objective function need to be manually adjusted. This makes multi-task learning very difficult and costly because it requires constantly trying various combinations of weights to determine the optimal task weights. In this paper, we propose a multi-dialect acoustic model that combines soft-parameter-sharing multi-task learning with Transformer, and introduce several auxiliary cross-attentions to enable the auxiliary task (dialect ID recognition) to provide dialect information for the multi-dialect speech recognition task. Furthermore, we use the adaptive cross-entropy loss function as the multi-task objective function, which automatically balances the learning of the multi-task model according to the loss proportion of each task during the training process. Therefore, the optimal weight combination can be found without any manual intervention. Finally, for the two tasks of multi-dialect (including low-resource dialect) speech recognition and dialect ID recognition, the experimental results show that, compared with single-dialect Transformer, single-task multi-dialect Transformer, and multi-task Transformer with hard parameter sharing, our method significantly reduces the average syllable error rate of Tibetan multi-dialect speech recognition and the character error rate of Chinese multi-dialect speech recognition.

## 1. Introduction

The rise of deep learning models improved the performance of dialect-dependent acoustic models (AM) [1,2,3,4,5], but the variability of dialects makes dialect-dependent AM reduce model performance when faced with multi-dialect data [6]. Therefore, how to create a unified multi-dialect AM is a problem worth exploring. After all, the cost of maintaining a multi-dialect AM is much lower than maintaining several single-dialect AMs. In this paper, we take the multi-dialects of Tibetan and Chinese as examples, and aim to build a multi-dialect speech recognition model based on multi-task learning (MTL) and Speech-Transformer [7].

Admittedly, MTL is not the only way to build multi-dialect speech recognition models. There are also methods of fine-tuning [8] or adding dialect information into the model [9] that can also improve the performance of multi-dialect speech recognition models. However, for low-resource languages such as Tibetan, MTL seems to be the most appropriate choice, because MTL is often used in low-resource languages [10,11,12,13,14,15]. 

When using MTL, there are two important issues to consider first: what kind of multi-task model structure to choose, and how to allocate the learning proportion according to the difficulty of each task in the multi-task objective function? The structure of the multi-task model can be divided into two types: hard-parameter-sharing and soft-parameter-sharing [16]. In the hard-parameter-sharing multi-task model, the hidden layer is a common layer for all tasks, and the top layer is an independent layer for each task. We used this approach in our previous work [17], which can improve the performance for both multi-dialect speech recognition and dialect ID/speaker identification. However, the hard-parameter-sharing method cannot learn both the task-specific features and the shared features between tasks, so it cannot reveal the relationship between different tasks and how they contribute to each other. For soft-parameter-sharing MTL, each task not only has its own module, but also has specific modules for sharing information among multi-task streams. Therefore, in the soft-parameter-sharing multi-task model, the information shared between tasks and the relationship between tasks can be determined through specific modules. The works of [18,19] use cross-stitch units or task-specific attention networks to learn shared features and task-specific features between tasks, which can show explicitly how a task provides the features of the layers and how they are weighted for another task. In this paper, we take dialect ID recognition as an auxiliary task and build a multi-stream model for the two tasks based on soft-parameter-sharing multi-task learning and Speech-Transformer. Not only that, several auxiliary cross-attention modules are also introduced between the two task streams to allow auxiliary tasks provide dialect information to the multi-dialect speech recognition task.

As for the task weight of the multi-task objective function, it can be adjusted manually or automatically [20]. Manual methods have a high cost due to the need to constantly try various weight combinations. In contrast, the idea of automatically adjusting the weights when training the model seems simple and easy [21,22,23,24,25]. These methods of automatically adjusting weights can be roughly divided into two categories: one based on gradients and the other based on loss values. For example, the work of [21] achieves task balance through gradient normalization, and the works of [22,23] constrains the learning degree of the model according to the change in the loss value of each training step. Given our AM is a multi-stream architecture, building a dynamic adaptive method from the gradient aspect only increases the burden on the model. Therefore, we introduce a loss proportion in the cross-entropy of each task for automatically adjusting multi-stream loss for each epoch learning.

The rest of the paper is organized as follows: Section 2 discusses the related work. In Section 3, we detail the multi-task transformer and adaptive cross-entropy loss. Experimental results and analysis are provided in Section 4. Finally, we describe our conclusions in Section 5.

## 2. Related Work

Due to the variability of dialects, speech recognition of multi-dialect languages is severely challenging. When a dialect-dependent acoustic model is tested on different dialects, its performance is greatly reduced. There have been several approaches to improve the performance of multi-dialect speech recognition models. One such approach is fine-tuning, where dialect-specific models are obtained simply by training a single model using all dialect data first and then fine-tuning it on dialect-specific data [8]. There are other approaches that provide the acoustic model with an auxiliary input, such as i-vectors [9] in output text, to make it adaptive to different dialects.

Multi-task learning is also the method used commonly to improve the performance of multi-dialect speech recognition, which is especially effective for low-resource languages. Hard-parameter-sharing of MTL is the most widely used approach, which learns a shared representation from a primary task and auxiliary tasks, and reduces the risk of overfitting on the original task. In the works of [26,27], for the auxiliary task of phoneme/dialect ID recognition and the primary task of speech recognition, a multi-dialect acoustic model is built on a hard-parameter-sharing multi-task structure, which improves the recognition accuracy of the auxiliary task and the primary task. In this approach, a common hidden layer for all tasks is predefined, and the shared features are not further filtered. Therefore, if the predefined general hidden layer is not studied carefully, the information transmitted between tasks become a disturbing factor [16]. In contrast, soft-parameter-sharing multi-task learning further processes the shared information through specific modules, such as the cross-stitch and attention modules in [18,19]. Moreover, in this multi-task structure, each task has its own model and parameters. Therefore, compared with hard parameter sharing, the soft-parameter-sharing MTL makes it easier to understand the process of task interaction. The work of [28] builds an ensemble of dialect-specific modules (or experts) for three English dialects. The outputs of the experts are then linearly combined using attention weights generated by a long short-term memory (LSTM) network. Experimental results show that their proposed model achieves an average word error rate reduction (WERR) of 4.74% compared to their baseline models.

Although the purpose of multi-task joint training is to improve the performance of the model on all tasks at the same time, it is often encountered that some tasks dominate other tasks during the training phase [22], which leads to improving the performance of only a few tasks and reducing the performance of other tasks. This dominance can be attributed to variations in task complexities, uncertainties, magnitudes of losses, etc. [29], and, therefore, an appropriate loss or prior strategy for some tasks in MTL is a necessity. There are several works that attempted to balance the task’s loss weights. The work of [22] proposed a principled approach to multi-task learning that weighs multiple loss functions by considering the homoscedastic uncertainty of each task. The work of [23] proposed dynamic weight average (DWA), which uses an average of task losses over time to weigh the task losses. The work of [24] proposed dynamic task prioritization, which uses the difficulty of tasks to adjust the task weights. This allows distributing focus on harder problems first and then on less challenging tasks. The work of [21] manipulates gradient norms over time to control the training dynamics, and the work of [25] proposed a multi-gradient descent algorithm from the perspective of gradient, but these methods are complicated and bring additional complexity to the training phase. In this paper, we take the adaptive cross-entropy loss function as the multi-task objective function, where the weight of each cross-entropy is dynamically adjusted according to the proportion of each task in the total loss.

## 3. Multi-Task Transformer

Our multi-task Transformer is a multi-stream multi-task network based on the Speech-Transformer [7]. In order to facilitate the following description, we first briefly introduce the components of the Speech-Transformer, and then introduce our proposed model in detail.

### 3.1. Speech-Transformer

#### 3.1.1. Convolutional Block

As shown in Figure 1, the Speech-Transformer consists of a convolutional block, an encoder, and a decoder. The convolutional block consists of two-layer 2D convolutional networks. Batch normalizations are employed around each of the convolutional layers. Considering that speech features are two-dimensional spectrograms with time and frequency axes, Speech-Transformer chooses a convolutional network to exploit the structural locality of spectrograms and mitigate length mismatches by striding along time.

#### 3.1.2. Encoder

Encoder is composed of a stack of N identical layers. Each layer has two sub-layers. The first sublayer is a multi-head self-attention mechanism, and the second one is a position-wise fully connected network. A residual connection is used after each of the two sub-layers. This is then followed by a layer normalization, which can normalize the same feature field so that the model has better robustness.

#### 3.1.3. Decoder

Decoder is similar to the encoder, except that a multi-head cross-attention is inserted to compute the correlation between the output of the encoder and the character features. During the training phase, in order to prevent early leakage of character features and maintain the autoregressive properties in the decoder, the self-attention sub-layer in the decoder masks all values corresponding to illegal connections.

### 3.2. Soft-MTL Transformer

Since our multi-task Transformer is a soft-parameter-sharing multi-stream structure, and the model of each task stream is based on Speech-Transformer, in the following section, our model is referred to as “Soft-MTL Transformer” for convenience of description.

As shown in Figure 2, the framework of the Soft-MTL Transformer has two streams, where the upper stream belongs to the dialect ID recognition task, and the lower stream belongs to the speech recognition task. Regardless of auxiliary cross-attention, the primary task stream is the same as Speech-Transformer, with 1 convolutional block, 12 identical encoders, and 6 identical decoders. The convolutional block and the encoder structure of the auxiliary task stream are also the same as the Speech-Transformer, including one convolutional block and six identical encoders. The decoder is replaced by a simple fully connected neural network with a soft-max layer because the auxiliary task is a simple classification task rather than a sequence-to-sequence problem.

To make the two task streams share information, we introduce auxiliary cross-attention into the model. The details of the auxiliary cross-attention are described as follows.

#### Auxiliary Cross-Attention

Further, to make the decoders of primary task stream take into account the dialect identification information for character sequence prediction, auxiliary cross-attention is inserted in each decoder of the primary task stream.

As shown in Figure 2, the auxiliary cross-attention in the decoder is located after the self-attention, and parallel with the original cross-attention in the second layer. The input of auxiliary cross-attention consists of the output of the self-attention layer ($Output_{self\text{-}attn}$), and the output of the encoder from the dialect recognition task ($Output_{auxiliary}$).

As shown in Figure 3, to calculate auxiliary cross-attention, the first step needs to create three vectors of Q, K, and V. These vectors are created through multiplying the two kinds of outputs by three weight matrices, $W^{Q}$, $W^{K}$, and $W^{V}$ learned during the training process. As shown in Formulas (1)–(3), $Output_{auxiliary}$ is used to create K and V, which represents the dialect information, and Q is created by $Output_{self\text{-}attn}$, which represents the text information.
(1)$$Q=Output_{self\text{-}attn}\cdot W^{Q}$$
(2)$$K=Output_{auxiliary}\cdot W^{K}$$
(3)$$V=Output_{auxiliary}\cdot W^{V}$$

We obtain the weights on V via Formula (4). The output of auxiliary cross-attention is referred to as ACA(Q,K,V).
(4)$$\mathrm{ACA}\left(Q,K,V\right)=\mathrm{Softmax}\left(\frac{Q\cdot K^{T}}{\sqrt{d_{k}}}\right)\cdot V$$

In Formula (4), $d_{k}$ is the dimension of Q and K, which is used to prevent the soft-max function from generating very small gradients. The auxiliary cross-attention performs in multi-head attention mechanism mode.

### 3.3. Adaptive Cross-Entropy Loss

The cross-entropy loss function is the most common loss function for training deep learning models due to its excellent convergence speed. The cross-entropy loss refers to the contrast between two random variables; it measures them in order to extract the difference in the information they contain. Here, during the training, we use this loss function for the primary task and auxiliary task to calculate the accuracy of the task model by defining the difference between the estimated probability and the desired outcome. Generally, the objective function of the multi-task model is the sum of the loss functions of each task, as shown in Formula (5).
(5)$$L_{multi\text{-}task}\left(t\right)=L_{primary}\left(t\right)+L_{auxiliary}\left(t\right)$$
where $L_{primary}$ is the primary task loss function and $L_{auxiliary}$ is the auxiliary task loss function, t is the training epoch. The detailed cross-entropy loss functions of $L_{primary}$ and $L_{auxiliary}$ at the $t^{th}$ training epoch are shown in Formulas (6) and (7), respectively.
(6)$$L_{primary}\left(t\right)=-\frac{1}{N}\left[\underset{\left(X,Y\right)\in Q}{\sum }\overset{L}{\underset{l=1}{\sum }}\log\left(P\left(y_{l}|y_{l-1},X\right)\right)\right]$$
(7)$$L_{auxiliary}\left(t\right)=-\frac{1}{N}\left[\underset{\left(X,Z\right)\in Q}{\sum }\overset{M}{\underset{m=1}{\sum }}\log(P(z_{m}|X)\right]$$

Here, Q represents the training data set with *N* samples, *X*, *Y*, and *Z* represent the speech frame sequence, character sequence, and dialect ID, respectively. L and M represent the vocabulary size and the number of dialects, respectively. 

For adaptively balancing multi-task learning, we introduce the loss proportions W as a factor in cross-entropy function. Therefore, $L_{primary}$ and $L_{auxiliary}$ are changed as shown in Formulas (8) and (9), respectively.
(8)$$L_{primary}\left(t\right)=-\frac{1}{N}\left[W_{primary}\left(t\right)\underset{\left(X,Y\right)\in Q}{\sum }\overset{L}{\underset{l=1}{\sum }}\log\left(P\left(y_{l}|y_{l-1},X\right)\right)\right]$$
(9)$$L_{auxiliary}\left(t\right)=-\frac{1}{N}\left[W_{auxiliary}\left(t\right)\underset{\left(X,Z\right)\in Q}{\sum }\overset{M}{\underset{m=1}{\sum }}\log(P(z_{m}|X)\right]$$
(10)$$W_{primary}\left(t\right)=\frac{L_{primary}\left(t-1\right)}{L_{primary}\left(t-1\right)+L_{auxiliary}\left(t-1\right)}$$
(11)$$W_{auxiliary}=\frac{L_{auxiliary}\left(t-1\right)}{L_{primary}\left(t-1\right)+L_{auxiliary}\left(t-1\right)}$$
(12)$$W_{primary}\left(t\right)+W_{auxiliary}\left(t\right)=1$$

As shown in Formulas (10) and (11), $W_{primary}$ and $W_{auxiliary}$ at the $t^{th}$ training epoch are determined by the proportion of the corresponding loss value at training epoch t−1 in the total loss. The proportion of a task reflects the learning status of the model in this task. The larger the proportion value, the more attention the task needs to pay. A task with the greater loss value at epoch t−1 has a greater proportion of the total loss. In the next training step, the model learns more about that task. 

In terms of speech recognition and dialect ID recognition tasks, the former is obviously more difficult than the latter, which means the loss proportion for the speech recognition task is greater than the one for the dialect ID recognition task. Therefore, combined with our dynamic adjustment strategy, our multi-task model appropriately adjusts the learning strategy according to the task difficulty and allocate appropriate attention to each task. Moreover, since our approach adds only two simple operations, there is no risk of increasing computational complexity.

In order to reduce the uncertainty caused by stochastic gradient descent and random training data selection, $L_{primary}$ and $L_{auxiliary}$ are calculated as the average loss value of each epoch over all iterations.

## 4. Experiment

### 4.1. Data

In the experiments, we evaluated our method using speech data from three Tibetan dialects and three Chinese dialects. Tibetan multi-dialect data come from the open data set TIBMD@MUC [30] and Chinese multi-dialect data come from Common Voice [31]. In this work, Tibetan syllables and Chinese characters were used as the recognition unit for Tibetan multi-dialect speech recognition and Chinese multi-dialect speech recognition, respectively. The embedding vector dimension of a Tibetan syllable or a Chinese character is 256, and the speech feature adopts the 40-dimensional Fbank feature. The detailed information of the data is shown in Table 1. The data of Ü-Tsang contain Lhasa and Rikaze sub-dialects, Amdo dialect data include Qinghai, Ngawa, and Xiahe sub-dialects. The data of Kham consist of Chamdo and Derge sub-dialects. For Chinese, we used the official Chinese language—Mandarin, Hong Kong’s Cantonese, and Taiwan’s Hokkien as Chinese multi-dialect data.

### 4.2. Settings

The convolutional block of each task stream contains two 2D CNN layers with stride 2, and kernel size 3. The channel number of the first layer is 16, and is 128 for the second layer.

The encoder in the primary task stream is a stack of 12 layers, however, it is 6 layers for the dialect ID recognition task stream. Each layer has 4 attention heads, and each head is a self-attention with 64 dimensions. The output of each attention head is concatenated and multiplied with a weighted matrix. Then a two-layer fully connected neural network with an activation function of “GLU” is used for nonlinear mapping. The node number is 2044 and 256 in these two layers, respectively.

The configuration of self-attention, cross-attention, and auxiliary cross-attention are 4 heads and 64 dimensions in decoders of the primary task stream.

The Soft-MTL Transformer is built on the Pytorch framework, and it is trained with 80 epochs. The batch size is 10. For optimization, the Adam algorithm [32] with gradient clipping is used. Its learning rate is 0.01. The label-smoothing technology [33] is used during training, and the smoothing parameter α = 0.1. The decoding process uses a beam search algorithm with 5 beam widths.

### 4.3. Experimental Results

For model evaluation, we designed three experiments. The first experiment compares the performance of the multi-task Transformer with soft parameter sharing proposed in this paper with the single-task Transformers and hard-parameter-sharing Transformer. The single-task Transformers are trained on dialect-specific data and the mixture of three dialects data, respectively. They are referred to as “Single-dialect Transformer” and “Multi-dialect Transformer”, respectively. The structure of both baseline models is the same as the standard Speech-Transformer. The hard-parameter-sharing Transformer is applied by sharing the convolutional blocks and encoders between two tasks, while keeping task-specific top layers, as shown in Figure 4. Except for the fact that there is no auxiliary cross-attention, the configuration of the other modules is the same as the Soft-MTL Transformer. “Hard-MTL Transformer” represents the MTL Transformer with hard parameter sharing.

In the second experiment, the method of dynamically adjusting the loss weights proposed in this paper is compared with the manual adjustment method, and also compared with the uncertainty weighting method [22] and DWA method [23]. “Uncertainty Weighting” refers to the method that dynamically assigns loss weights according to the uncertainty of each task. “DWA” represents the method that averages loss weighting over time by considering the rate of change of loss for each task. Since the multi-task Transformer with adaptive cross-entropy proposed in this paper is a soft-parameter-sharing multi-task structure, other methods that are only suitable for hard-parameter-sharing multi-task models [21] or have high computational complexity [24,25] are not compared.

The last experiment is about identifying dialect IDs. In this experiment, we not only compare the performance of two different types of multi-task models on auxiliary tasks, but also investigate the performance of “Soft-MTL Transformer” with the ability to dynamically adjust loss weights on auxiliary tasks.

For each model in all experiments, we assigned equal budgets to optimization, learning rate, and the number of training epochs to ensure fair comparisons.

#### 4.3.1. Single-Task Transformer vs. Multi-Task Transformer

We adopted the syllable error rate (SER) for Tibetan multi-dialect speech recognition and the character error rate (CER) for Chinese multi-dialect speech recognition to evaluate the performance of the model. The loss proportions $W_{primary}$ and $W_{auxiliary}$ in cross-entropy loss of Hard-MTL Transformer and Soft-MTL Transformer were manually assigned as 0.9 and 0.1, respectively. The results in Table 2 are the best recognition results of the four models.

Obviously, “Single-dialect Transformer” has the highest average recognition error rate for three dialects in the Chinese and Tibetan test sets. “Multi-dialect Transformer” outperforms the “Single-dialect Transformer” in all dialects. This shows that the three dialects of each language have commonalities, and the “Multi-dialect Transformer” trained on the joint data of multiple dialects promotes the model to share parameters among dialects.

Compared with the “Single-dialect Transformer”, our “Soft-MTL Transformer” has lower CERs and SERs for all dialects. This shows that the multi-dialect model constructed by our multi-task learning method is effective in both Chinese and Tibetan languages. Compared with the “Multi-dialect Transformer”, our “Soft-MTL Transformer” still has better results on Mandarin and Cantonese for Chinese, and Ü-Tsang and Amdo dialects for Tibetan. But unfortunately, the CER for the Hokkien dialect of Chinese and the SER for the Kham dialect of Tibetan increases.

Clearly, among these multi-dialect speech recognition models, our proposed model has an excellent performance for both languages, and achieves 11.47%, 2.52%, and 6.41% lower than the “Hard-MTL Transformer” on the CERs for Mandarin, Cantonese, and Hokkien, respectively. For the SERs of the Ü-Tsang, Amdo, and Kham dialects of Tibetan, our model is also lower than the “Hard-MTL Transformer”, at about 1.67%, 1.4%, and 11.24%, respectively.

#### 4.3.2. Adaptive Cross-Entropy Loss Experiment

Before using the dynamic adjustment method, we tested the performance of the Soft-MTL Transformer on two tasks of Tibetan multi-dialect by manually using various combinations of task weights. As shown in Figure 5, there is a positive correlation between the performance of the model in each task and their weights. In other words, the greater the task weight, the better the performance of the model in the corresponding task. The primary task is the Tibetan multi-dialect speech recognition task, which has the lowest SER at a task weight of 0.9. Although the auxiliary task is not as sensitive to the task weight as the primary task, its performance also tends to improve with the increase in task weight.

The sensitivity of the model to the task weights verifies the necessity of dynamically adjusting task weights. The performance of the models in the speech recognition task is listed in Table 3.

It is apparent that the CERs and SERs of the “Uncertainty weighting” and “DWA” methods do not perform better than manual adjustments for the three dialects of the two languages. It shows that, although these two methods are easy to implement, their dynamic adjustment mode is not suitable for our model. Surprisingly, our method “Adaptive Soft-MTL Transformer” has better results than the manual adjustment method in the Chinese and Tibetan test sets, and the average CERs and SERs for both language data sets decrease by 0.62% and 5.48%, respectively. Compared with the “Single-dialect Transformer”, the “Adaptive Soft-MTL Transformer” model combined with our adaptive cross-entropy loss method reduces the average CER and SER by 34.57% and 32.24% on the Chinese and Tibetan test sets, respectively. Compared with the “Multi-dialect Transformer”, it also decreases by 1.61% and 2.7%, respectively.

#### 4.3.3. Dialect ID Recognition Experiment

Multi-task learning refers to the ability of the model to learn multiple tasks at the same time. It is not only as simple as sharing information between tasks, but also needs to improve the performance of all tasks. After the above experiments, it can be proved that the method of multi-task Transformer with adaptive cross-entropy loss (denoted as “Adaptive Soft-MTL Transformer”) has good performance in multi-dialect speech recognition. In this experiment, we evaluate the dialect ID recognition accuracy, and then compare the “Adaptive Soft-MTL Transformer” with the “Hard-MTL Transformer” and “Soft-MTL Transformer”; Table 4 shows the experimental results.

“Hard MTL Transformer” and “Soft MTL Transformer” are models that manually adjust the weight of loss function. Table 4 presents the dialect ID recognition results when these two models perform best in speech recognition and dialect ID recognition, and at that time, the weight of auxiliary tasks is 0.1. Comparing the dialect ID recognition results of these two models, it can be seen that, except for the low-resource dialects of the two languages (Hokkien and Kham), “Soft-MTL Transformer” has higher accuracy than “Hard-MTL Transformer”, and the average accuracy of “Soft-MTL Transformer” for the two languages is 99.17% and 99.74%, which are 0.13% and 0.31% higher than that of “Hard-MTL Transformer”, respectively. This proves that the “Soft-MTL Transformer” also has better performance than the “Hard-MTL Transformer” in auxiliary tasks.

In addition, comparing “Soft-MTL Transformer” and “Adaptive Soft-MTL Transformer”, it can be seen that “Adaptive Soft-MTL Transformer” further improves the accuracy of dialect ID recognition for the Tibetan multi-dialect data set. However, for Chinese multi-dialect data, the performance of “Adaptive Soft-MTL Transformer” slightly declines, and the average accuracy is 0.25% lower than that of the “Hard-MTL Transformer”. Although our method declines in Chinese dialect ID recognition, the overall average accuracy rate remains around 98.79%, with high recognition accuracy.

## 5. Conclusions

In this paper, we propose an adaptive multi-task Transformer with multiple streams that operate on Chinese and Tibetan multi-dialect speech recognition. The Transformer with soft parameter sharing provides the information between the primary task and auxiliary task through auxiliary cross-attention in the decoder of the model. We demonstrate that the way in which dialect information is shared by auxiliary cross-attention for the decoder of the primary task effectively improves the multi-dialect speech recognition ability of the model. Moreover, we also propose an adaptive cross-entropy loss to adjust the task-learning proportion in the multi-task objective function. Experiments show that our method not only outperforms methods that manually adjust task weights, but also achieves better results than other dynamic adjustment strategies.

## Figures and Tables

**Figure 1 entropy-24-01429-f001:**
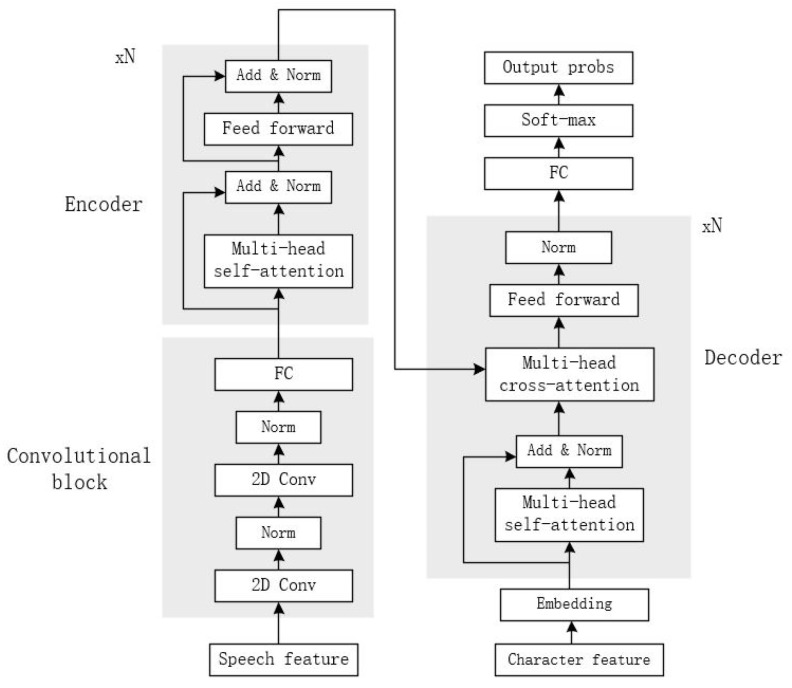
The structure of Speech-Transformer.

**Figure 2 entropy-24-01429-f002:**
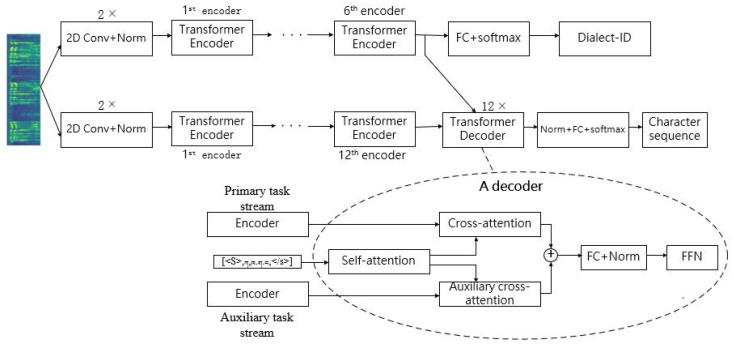
The structure of Soft-MTL Transformer.

**Figure 3 entropy-24-01429-f003:**
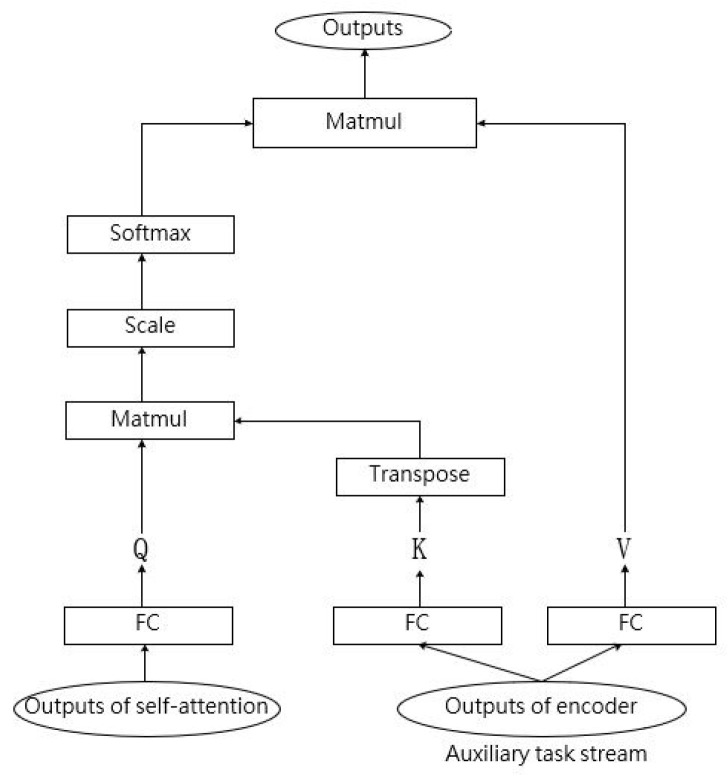
One head of auxiliary cross-attention.

**Figure 4 entropy-24-01429-f004:**
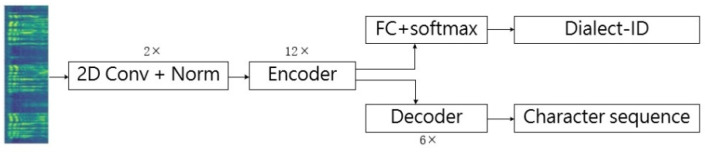
The structure of Hard-MTL Transformer.

**Figure 5 entropy-24-01429-f005:**
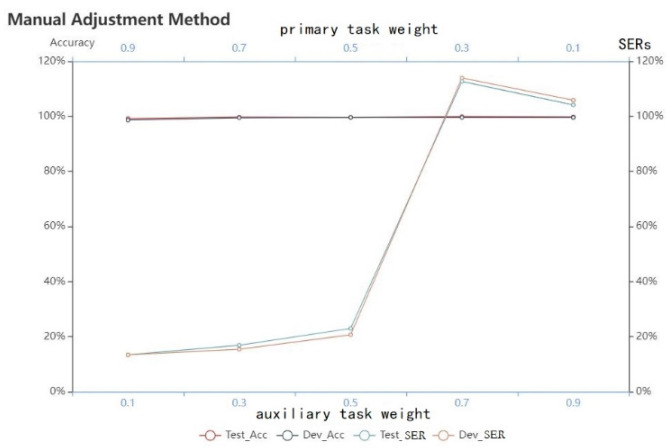
Comparison of task weights for Soft-MTL Transformer.

**Table 1 entropy-24-01429-t001:** Data information.

Language	Dialect	Training Data (Hours)	Training Utterances	Test Data (Hours)	Test Utterances
Tibetan	Ü-Tsang	23.74	20,594	2.98	2575
	Amdo	18.93	17,286	2.36	2161
	Kham	2.61	2373	0.33	269
Chinese	Mandarin	34.66	23,755	2.46	1126
	Cantonese	12.96	6695	1.63	961
	Hokkien	7.74	9967	0.69	1074

**Table 2 entropy-24-01429-t002:** The performance of single-task models and multi-task models.

Model	CER(%)			SER(%)		
	Chinese Test Data			Tibetan Data		
	Mandarin	Cantonese	Hokkien	Ü-Tsang	Amdo	Kham
Single-dialect Transformer	32.72	63.20	94.43	14.96	8.90	107.73
Multi-dialect Transformer	25.44	13.05	**52.97**	8.96	4.26	**24.33**
Hard-MTL Transformer	34.78	13.96	60.15	9.53	4.74	40.64
Soft-MTL Transformer	**23.31**	**11.44**	53.74	**7.86**	**3.07**	29.40

**Table 3 entropy-24-01429-t003:** Experimental results of dynamically adjusting loss weights method.

	CER(%)			SER(%)		
	Chinese Test Data			Tibetan Test Data		
	Mandarin	Cantonese	Hokkien	Ü-Tsang	Amdo	Kham
Manually weighting	**23.31**	11.44	53.74	7.86	3.07	29.40
Uncertainty weighting [22]	48.13	32.49	66.03	12.88	8.51	42.27
DWA [23]	46.31	24.16	54.37	12.06	6.67	44.22
Adaptive Soft-MTL Transformer	24.06	**11.22**	**51.36**	**5.67**	**2.70**	**26.48**

**Table 4 entropy-24-01429-t004:** Experimental results of dialect ID recognition.

Model	Recognition Accuracy(%)					
	Chinese Test Data			Tibetan Test Data		
	Mandarin	Cantonese	Hokkien	Ü-Tsang	Amdo	Kham
Hard-MTL Transformer	99.73	99.19	**98.21**	99.38	99.58	99.32
Soft-MTL Transformer	**99.93**	**99.71**	97.87	99.73	99.86	96.62
Adaptive Soft-MTL Transformer	99.73	99.34	97.31	**99.85**	**99.91**	**100.00**

## Data Availability

The data supporting the conclusions of this paper is available at http://www.openslr.org/124/ (accessed on 29 June 2022) and Common Voice (https://commonvoice.mozilla.org/en/datasets/ (accessed on 29 June 2022)).

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
