# Peer review of "Multi-Task Transformer with Adaptive Cross-Entropy Loss for Multi-Dialect Speech Recognition"

_entropy, 2022, doi:10.3390/e24101429_

Round 1

Reviewer 1 Report

This work applies a current technique to dialects that have not received enough attention in the literature. The positive results of both identifying the dialect and the recognition results at the same time might encourage other researchers to apply this at a larger scale. (I'm not familiar enough with the current literature to say for certain if this is particularly novel, but the approach and experiments are well presented.)

There are numerous grammatical mistakes, I'm giving a partial list of things that I noted. L9 'make -> makes it', L10 Meanwhile is the wrong word choice, L12: 'it' pronoun has unclear reference, L38: fine-tuning or adding, L73 this paragraph is repeated (largely) in the related work section, L86: severely challeng_ing_. When _a_ L100 'the common' -> a common, L104: no the before soft parameter, L120: insert 'that have' between works and attempterd L141 the -> an and a.

The unanswered question of how to pick the task weight for dialect might be related to the question about how far apart these dialects are with respect to each other. In fact the dialect ID results are not evident in the paper, and it would be better if these results were summarized as well.

Author Response

Dear reviewer,
       Thanks for your comments on our manuscript entitled “Multi-task Transformer with Adaptive Cross-entropy Loss for Multi-dialect Speech Recognition” (Manuscript ID: entropy-1879418). Those comments are all valuable and helpful for revising and improving our paper. We have studied all your comments carefully and have made conscientious correction. Revised portion are marked up using the “Track Changed” function in the paper. The main corrections in the paper and the responses to your comments are as follows:

Comments1: There are numerous grammatical mistakes, I'm giving a partial list of things that I noted. L9 'make -> makes it', L10 Meanwhile is the wrong word choice, L12: 'it' pronoun has unclear reference, L38: fine-tuning or adding, L73 this paragraph is repeated (largely) in the related work section, L86: severely challeng_ing_. When _a_ L100 'the common' -> a common, L104: no the before soft parameter, L120: insert 'that have' between works and attempterd L141 the -> an and a.
Respoonse1: All the grammatical errors you pointed out have been fixed, not only that, but the full paper has been re-checked and other potential problems have been fixed.
Comments2: In fact, the dialect ID results are not evident in the paper, and it would be better if these results were summarized as well.
Response2: We added the recognition results of dialect IDs in Sec. 4.3.3 in the revised version, and also give the detailed analysis and conclusions.

       We greatly appreciate your comments on our paper. looking forward to hearing from you.

Sincerely,
Zhengjia Dan

Reviewer 2 Report

This paper proposes a novel method of multi-task Transformer combined with automatic adjustment for multi-dialect speech recognition. The core idea of the multi-task Transformer is auxiliary cross-attention. The auxiliary cross-attention processes the dialect information via the auxiliary task by the attention algorithm, and then provides it to the primary task so that the primary task discriminate dialects. In Table 2, "Soft-MTL Transformer" has a lower CER or SER than "Single-dialect Transformer" and "Multi-dialect Transformer" on both Tibetan dialects and Chinese dialects. Furthermore, the "Soft-MTL Transformer" also obtains better performance than the "Hard-MTL Transformer".

Another advantage of this paper is that it proposes a novel approach to automatically adjust the weights of the two cross-entropy loss functions according to the loss proportion at each iteration. This method does not increase computational complexity compared to manual adjustment and gradient-based adjustment methods.

However, I still have some concerns about the experiments. In the experiments, the paper lacks to provide the results on dialect ID recognition for multi-task model, which is mentioned in the motivation as another important task to prove the effectiveness of the proposed model.

Author Response

Dear reviewer,
        Thanks for your comments on our manuscript entitled “Multi-task Transformer with Adaptive Cross-entropy Loss for Multi-dialect Speech Recognition”. (ID: entropy-1879418). Those comments are all valuable and helpful for revising and improving our paper. We have studied all your comments carefully and have made conscientious correction. Revised portion are marked up using the “Track Changed” function in the paper. The main corrections in the paper and the responses to your comments are as follows:

Comment1: In the experiments, the paper lacks to provide the results on dialect ID recognition for multi-task model, which is mentioned in the motivation as another important task to prove the effectiveness of the proposed model.
Response: We added the recognition results of dialect IDs in Sec. 4.3.3 in the revised version, and also give the detailed analysis and conclusions.

      We greatly appreciate your comments on our paper. looking forward to hearing from you.

Sincerely,
Zhengjia Dan
